

**Global and regional phosphorus budgets in agricultural systems and their**
**implications for phosphorus-use efficiency**
Fei LUN [1], Junguo LIU [2]*, Philippe CIAIS [3], Thomas NESME [4], Jinfeng CHANG [3],
Rong WANG [3], Daniel GOLL [3], Jordi SARDANS [5], Josep PEÑUELAS [6], Michael
OBERSTEINER [7]
1. College of Resources and Environment Sciences, China Agricultural University,
Beijing 100193
2. School of Environmental Science and Engineering, South University of Science and
Technology of China, Shenzhen, 518055, China
3. IPSL – LSCE, CEA CNRS UVSQ, Centre d'Etudes Orme des Merisiers, 91191 Gif-
sur-Yvette, France
4. Bordeaux Sciences Agro, Université de Bordeaux, UMR 1391 ISPA, CS 40201,
33175 Gradignan Cedex, France
5. CREAF, Cerdanyola del Vallès 08193, Catalonia, Spain
6. CSIC, Global Ecology Unit CREAF-CSIC-UAB, Cerdanyola del Vallès 08193,
Catalonia, Spain
7. International Institute for Applied Systems Analysis, 2361 Laxenburg, Austria
* Author for correspondence: Junguo Liu. Phone/fax: +86-0755-88018012. Email:
liu_junguo@163.com
**Keywords**: phosphorus budget, phosphorus-use efficiency, global scale, regional scale,
country scale, agriculture



**Abstract**
The application of phosphorus (P) fertilizer to agricultural soils increased by 3.2%
annually from 2002 to 2010. We quantified in detail the P inputs and outputs of
cropland and pasture, and the P fluxes through human and livestock consumers of
agricultural products, at global, regional, and national scales from 2002 to 2010.
Globally, half of the total P input (21.3 Tg P yr$^{-1}$) into agricultural systems accumulated
in agricultural soils during this period, with the rest lost to bodies of water through
complex flows. Global P accumulation in agricultural soil increased from 2002 to 2010,
despite decreases in 2008 and 2009, and the P accumulation occurred primarily in
cropland. Despite the global increase of soil P, 32% of the world's cropland and 43%
of the pasture had soil P deficits. Increasing soil P deficits were found for African
cropland, versus increasing P accumulation in Eastern Asia. European and North
American pasture had a soil P deficit because continuous removal of biomass P by
grazing exceeded P inputs. International trade played a significant role in P
redistribution among countries through the flows of P in fertilizer and food among
countries. Based on country-scale budgets and trends we propose policy options to
potentially mitigate regional P imbalances in agricultural soils, particularly by
optimizing the use of phosphate fertilizer and recycling of waste P. The trend of
increasing consumption of livestock products will require more P inputs to the
agricultural system, implying a low P-use efficiency aggravating the P stocks scarcity
in the future. The global and regional phosphorus budgets and their PUEs in agricultural
systems is publicly available at https://doi.pangaea.de/10.1594/PANGAEA.875296.



## 1. Introduction

Population increases and dietary changes require higher food production, which increases global demand for fertilizers (Grote *et al*., 2005; Foley *et al*., 2011). Phosphorus (P) is an essential element for all organisms, and a lack of P limits growth. Fertilizer P enhances agricultural production, but P is also fixed in soils and can accumulate. In countries with high fertilizer use, much P is lost to leaching and runoff, leading to eutrophication of both inland and coastal waters (Carpenter *et al*., 1998; MacDonald *et al*., 2011).

To supply the growing need for P in fertilizer, mining of phosphate rock has quadrupled in the past half century, increasing from 46 Mt in 1961 to 198 Mt in 2011 (Scholz *et al*., 2013). Despite some short-term fluctuations in the price of phosphate rock, the global production of fertilizer P has been steadily increasing, at a rate of 3% to 4% annually during the half century before 2011, and is projected to increase by 50 to 100% by 2050 (Cordell *et al*., 2009, 2012). Extractable phosphate rock is a non-renewable resource, and significant depletion of the resource is projected by the end of this century if the current intensive use continues, possibly leading to resource shortages (Cordell *et al.*, 2009; van Vuuren *et al.*, 2010; Peñuelas *et al.*, 2013).

The mining of P and its application as fertilizer in cultivated land is a major anthropogenic perturbation of the natural biogeochemical P cycle (Carpenter and Bennett, 2011; Elser and Bennett, 2011; Steffen *et al*., 2015). The negative impacts of this perturbation on the natural environment depend on how much P is lost from regions with intensive fertilizer use (Smil, 2000; Bennett *et al*., 2001).

P application differs significantly between countries and crop types (Grote *et al*., 2005), and previous researchers have attempted to estimate the P flows in agricultural systems in Europe (Ott & Rechberger, 2012), the United States (Suh & Yee, 2011),



China (Ma *et al*., 2011), France (Senthilkumar *et al*., 2012), Australia (Cordell *et al*.,
2013), and the world (Smil, 2000; Liu *et al*., 2008; MacDonald *et al*., 2011; Schipanski
& Bennett, 2012). International trade and regional agricultural policies affect P budgets
by increasing or decreasing the gap between P inputs and P outputs in agricultural land
(Grote *et al*., 2005). Previous research mainly focused on cropland while P fluxes in
pasture and livestock production systems received less attention (McDowell and
Condron, 2004) hampering the differences in methodologies, system boundaries, and
data sources have made it difficult to assess the differences in the phosphorus use
efficiencies among agricultural sectors and to extrapolate regional findings to the global
scale.
To mitigate these problems, we (1) compiled a detailed and harmonized dataset of
P fluxes in agriculture for countries around the world, including detailed analysis of
input and output fluxes for cropland, managed grassland (hereafter, pasture), livestock,
and human consumers of agricultural products; (2) characterized P budgets and P-use
efficiencies in those different sub-systems; and (3) examined how international trade of
phosphate fertilizer and agricultural commodities influences regional P fluxes. We
performed this analysis at the scale of countries, regions, and the world; wherever
possible, we distinguished different crop types. The study period was from 2002 to
2010, allowing us to study temporal trends.
**2. Materials and methods**
In this study, we obtained data for 224 countries (Table SI-1 in the supporting
information). We defined the agriculture system as cropland and pasture ecosystems,
plus human and livestock consumers of agricultural production and of other products
containing P (Fig. 1). External P inputs to the agriculture system came from mined
phosphate rock and atmospheric deposition. Several processes cause P losses from the





system into the external environment (here, defined as non-agricultural land and bodies
of water). Figure 1 presents the fluxes of P into and out of the agriculture system at a
global scale, including internal fluxes between ecosystems and consumers. We
quantified these fluxes in the present study based on a mass-balance approach (Cordell
*et al*., 2012). We defined the phosphorus-use efficiency (PUE) of the agricultural
system and of its subsystems as the ratio of the total P harvested in economic outputs
(e.g., crops, meat, milk and eggs) to the total P input.  International trade in fertilizer
and food is discussed separately in section 2.3. The data sources and an overview of the
mass-balance equations are presented in the rest of this section; details and equations
are presented in the Supporting Information (SI).
**2.1 P flows into and out of the agricultural system**

Inputs into the agricultural system, which is within the gray box in Fig. 1, are from

mined phosphate rocks and atmospheric deposition. We did not include P from in situ
weathering of soil particles because the rate of this process is insignificant compared
with the magnitude of other inputs (Liu *et al.*, 2008). Outputs included P emission into
the atmosphere from fires and P loss to uncultivated land or bodies of water.

**2.1.1 P inputs**

Data on agricultural inputs of phosphate P in fertilizers were collected from the

International Fertilizer Industry Association (http://www.fertilizer.org) and divided
between cropland and pasture uses based on information from FAO (2002) and the
FAOSTAT database (http://www.fao.org/faostat/en/#data). A small fraction (8%) of P
from mined phosphate rock is used to produce animal feed additives. Apart from
fertilizer and animal feed additives, the rest of the mined P is used to produce detergents
and other products directly consumed by humans (Ringeval *et al*., 2014). Atmospheric
P deposition in cropland and pasture areas was calculated separately in each country





using gridded global P-deposition maps obtained using the LMDz-INCA aerosol
chemistry transport model of Wang *et al*. (2014, 2015) and agricultural land-use maps.
Details are provided in the Supporting Information (Table SI-2).

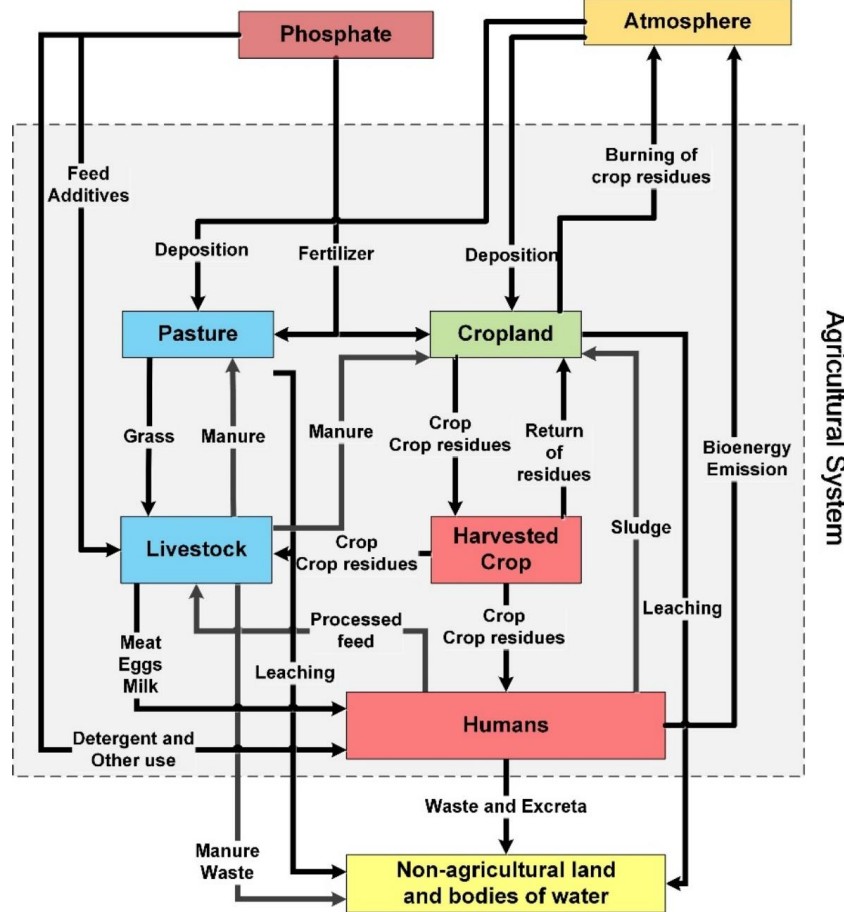


Figure 1: Scheme of the P pools and fluxes used to diagnose global P budgets for the
agricultural sector. The agricultural sector (or system) in the grey box includes cropland
and pasture soils, livestock, human consumers of livestock and crop products and users
of phosphate derived products. National and regional P budgets are calculated using the
same scheme, but including in addition exports and imports of P embedded in traded
crop and livestock products, and fertilizers.



**2.1.2 P outputs**


P emission from agricultural fires was obtained from the gridded dataset of Wang
*et al.* (2015), and cover the burning of crop residues in the field, by households, and for
the production of bioenergy from crop biomass. Leaching from cropland and pasture
soils was assumed to be a constant fraction (12.5%) of P inputs for each agricultural
land use type (Bouwman *et al.* 2013). P outputs from non-recycled livestock and human
manure were calculated based on the mass balance. Note that erosion-induced losses of
P are important in many agricultural regions (Quinton *et al.*, 2010), but were not
considered in this study because we lack data on the re-deposition of P in eroded soil
material from agricultural soils. In future research, it will be important to quantify this
source of P, particularly in agricultural areas that receive large annual inputs of
sediment (e.g., in river floodplains and sites on steep terrain that experience significant
erosion farther up the slope followed by deposition).
**2.2 P flows within the agricultural system**
**2.2.1 P in harvested crop biomass and crop residues**
The flux of P in harvested crop biomass was estimated from yield data (FAOSTAT)
using crop-specific P concentrations, after grouping 178 different crops into 13 crop
types (COMIFER, 2007; USDA-NRCS, 2009; Waller, 2010; Table SI-2). P in
harvested crop biomass was partitioned into crops (for human and livestock
consumption) and crop residues (Fig. 1). We estimated the P fluxes of crop residues
from FAOSTAT data and from Liu *et al.* (2008) to account for residue that is recycled
in the field (50%), transformed into livestock feed (25%), and burned or used by other
human activities (25%).
**2.2.2 P in grazed biomass**
The P removed from pasture by livestock grazing was estimated by combining



forage grass consumption data with the P concentrations in grass biomass (Antikainen
*et al.*, 2005; COMIFER, 2007; USDA-NRCS, 2009; Waller, 2010). Gridded data on
grass biomass consumption by livestock were obtained by combining the global
livestock production systems dataset of Herrero *et al.* (2013) with pasture net primary
productivity simulated by the ORCHIDEE-GM global pasture model (Chang *et al.*,
2013, 2015). We chose the ORCHIDEE-GM model for this analysis because it is able
to separate the intake of grazed vs. cut forage grass.

### 2.2.3 P in animal feed products

Animal feed products used as complementary diet ("feed additives") represent
direct inputs to the livestock sub-system (Fig. 1). This flux was deduced from the mass
balance of the known input and output fluxes for the livestock P pool, but did not
account for long-term changes in P storage in that pool. See the Supporting Information
for more details.

### 2.2.4 P embedded in livestock products

This flux of P leaving the livestock subsystem and entering the human subsystem
(Fig. 1) through the harvesting of products was calculated by multiplying the
FAOSTAT production data for meat, eggs, and milk by the product-specific P
concentrations reported by Grote *et al.* (2005).

### 2.2.5 P in livestock manure

We calculated the manure P production based on FAOSTAT data about N in
livestock manure and P:N values for each types of livestock manure (MWPS-18, 1985;
OECD Secretariat, 1991; Levington Agriculture, 1997; Sheldrick et al., 2003; ASAE,
2005) (see in the Table SI-3). Once produced, manure P is either applied to cropland,
left in the pasture, or lost to the environment as waste (Fig. 1), following the same
partitioning as that for N in the manure from FAOSTAT.



**2.2.6 P in human sewage sludge**

We assumed that the P output from humans equaled the inputs from non-fertilizer

P ore products and the consumed crop and livestock products (Fig. 1), and used this to
calculate the total P production in human excreta. P in human sewage sludge was
estimated using population data and values of per capita production of P in excreta
(Smil, 2000; Cordell *et al*., 2009). Following the method of Liu *et al*. (2008), we
assumed that 30% of the excreta P from urban populations and 70% of P from rural
populations were returned to cropland, either directly or after treatment of sewage
sludge, with the remaining P assumed to be lost to the environment (e.g., in landfills or
bodies of water).
**2.3 P flows from international trade**

We compiled the flows of P in international trade both from the P embodied in

crops and livestock products and in P embodied in fertilizers exchanged between
countries. For agricultural commodities, we used FAOSTAT data that provided a
matrix of commodities exchanged between countries, and converted this data into P
fluxes using commodity-specific P content data. For P fertilizers, we used the
International Fertilizer Industry Association trade statistics. By convention, a positive
trade balance for a country means that it is a net P importer. In addition, P fluxes
associated with the international trade of fertilizers, food, feed, and fiber commodities
can be associated with local cropland PUE and pasture PUE. We defined the
dependency on fertilizer imports ($F_{fer}$) as the ratio of the P in imported fertilizers ($P_{fer-imp}$)
to the P in all fertilizers consumed by a country ($P_{fer-con}$). Similarly, we defined the
dependency on food imports ($F_{food}$) as the ratio of P in food imports ($P_{food-imp}$) to the P
in all food consumed by a country. Furthermore, we defined $F_{total}$ as the ratio of the
total P imported (food and fertilizers) to the total P consumed as fertilizers and food in



a country. The equations for these calculations are presented in sections 2 to 6 of the
Supporting Information.
**2.4 Annual P budgets of cropland and pasture soils**
Annual changes in P stocks in cropland and pasture soils ($\Delta P$) were estimated as
the difference between inputs and outputs (i.e., the budget); $\Delta P > 0$ indicates net P
accumulation in the soil, $\Delta P < 0$ indicates a net deficit, and $\Delta P = 0$ represents no net
change. $\Delta P$ calculated in this manner does not reflect the legacy effects from previous
management and fertilization practices (Ringeval *et al*., 2014), but it is a useful metric
to identify regions with a P surplus or deficit at any point in time and to compare
countries.
Annual soil $\Delta P$ values were calculated as the differences between annual inputs
and outputs. Details and the equations are presented in section 2 of the Supporting
Information.
**2.5 Cumulative P budgets of cropland and pasture soils**
Following the method of Sattari *et al*. (2012), we separated the P inputs to soils
(except inputs in seeds) into two pools: (1) a stable P pool, which represents P that is
unavailable to plants on an annual basis, such as the P absorbed onto iron and aluminum
oxides (20% of total P inputs, including fertilizers, manure, sludge and deposition); and
(2) a labile P pool that is assumed to be available for plant uptake (80% of total P inputs).
P can be exchanged between the two pools. If inputs of labile P are larger than P
removal in crop biomass, we assumed that the surplus labile P gets transferred into the
stable P pool at the end of the year. In the opposite case, in which inputs of labile P are
lower than P removal, plants can take up P from the stable pool (Sattari *et al*., 2012).
This approach assumes that the P loss by runoff and leaching into bodies of water is
from the labile P pool only, and that P stored in seeds does not belong to either the



stable pool or the labile pool. This approach is simplistic, as more research will be
required to allow a more realistic modeling of these two pools and of the flows they are
involved in.

### 2.6 Phosphorus-use efficiency

We defined PUE as the ratio of P in the harvested economic outputs to P in the
inputs for the entire agricultural system (the gray area in Fig. 1) or for a given subsystem.
PUE indicates how much of the input P is transferred into value-added products. If
PUE > 1, the input of P is insufficient to sustain the output (harvested P), suggesting a
net reduction of the system's P reservoir. For cropland PUE, we defined P in harvested
crops as the economic P output of the crops, and the sum of phosphate fertilizer,
livestock manure, human sewage sludge, and P from atmospheric deposition as the P
input. For pasture PUE, harvested P refers to the P consumed by grazing animals, and
the sum of phosphate fertilizer, livestock manure going to the pasture, and P from
atmospheric deposition as the total inputs. For the livestock subsystem, the harvested P
output represents the P in livestock products (meat, eggs, and milk), whereas the inputs
represent the input into livestock. We also defined the PUE of human food ($\varepsilon_{food}$) as the
ratio of the P content in human excreta to the total P input in human food; this represents
an inconsistency with our previous definitions, since human excreta have currently no
economic value. The equations for all the PUE terms are provided in section 5 of the
Supporting Information.

### 2.7 Uncertainty estimates

Uncertainties in each flux originate both from the material flux data and from data
on the P concentration in each material considered by our analysis, including crop
products, crop residues, livestock, meat, eggs, milk, livestock, and human excreta.
Many of the global statistical datasets used in our analysis are not replicated, and no



alternative dataset is available for establishing a range of uncertainty values for the
different P fluxes. National datasets have usually not been formally analyzed to
determine their uncertainty, and many of the sources of uncertainty are difficult to trace
(e.g., clerical errors, differences between countries in product definitions). Thus, we
have only addressed the effect of uncertainties in the P concentration by means of
Monte Carlo simulations (3000 iterations) using the range of P concentrations reported
in the literature (Table SI-5).
**3. Results**
**3.1 Global agricultural P flows and their trends**
**3.1.1 Global P fluxes in and out of the agricultural system**
Figure 2 summarizes the annual average of global P flows for the period from 2002
to 2010. P from phosphate fertilizers was the largest single input flux, representing 93%
of the 21.3 Tg P yr$^{-1}$ of the global input, and most of it (82.4%) goes to cropland and
pasture. Outputs from the agriculture system amounted to 12.5 Tg P yr$^{-1}$, which
combines outputs from leaching and runoff into bodies of water (5.4), non-recycled
manure waste (4.3) and sewage (2.2), bio-energy (0.4), and burned crop residues (0.2).
The global annual P balance of agricultural systems was therefore positive during the
entire study period, with 8.8 Tg P yr$^{-1}$ accumulating in soil, of which 6.6 Tg P yr$^{-1}$
accumulated in cropland and 2.2 Tg P yr$^{-1}$ in pasture. On average, 41% of the P input
accumulated in soils from 2002 to 2010.





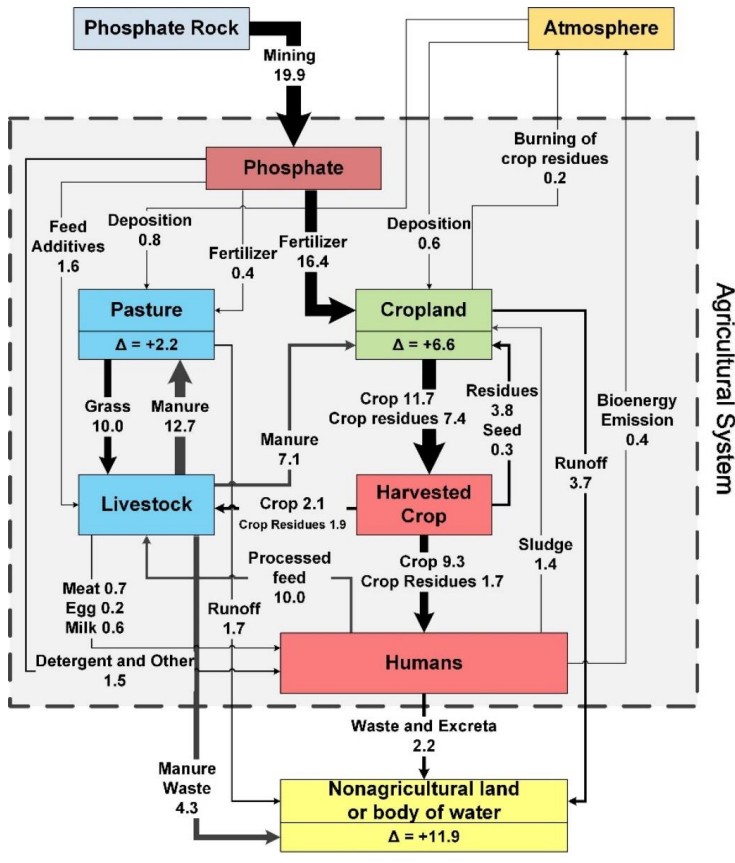

Figure 2: Annual P flows in the global agriculture system from 2002 to 2010. Values
are Tg P yr$^{-1}$. The notation Δ denotes the average change of P in pasture and cropland
soils, respectively. By convention, a positive value means accumulation. Note that
livestock and humans changes of P are assumed to be zero.
**3.1.2 Temporal trends**
Figure 3 shows the trends for the four largest P fluxes in the agriculture system.
Application of phosphate fertilizer increased at an average annual rate of 3.2% from
2002 to 2010, despite a decrease in 2008 that reflected reduced fertilizer application at
a time when the price of phosphate fertilizers increased (Cordell *et al*., 2009, 2012).
The trend for P in harvested crop biomass was also a steady increase, but at a lower
annual rate (2.4%) and with no decrease in 2008, probably because of the availability



of P that accumulated in the soil from previous years (as described in section 2.5).
Overall, P in agricultural soils increased by 1.3% annually, whereas P losses to the
environment increased faster (6.4% $yr^{-1}$) than fertilizer inputs.

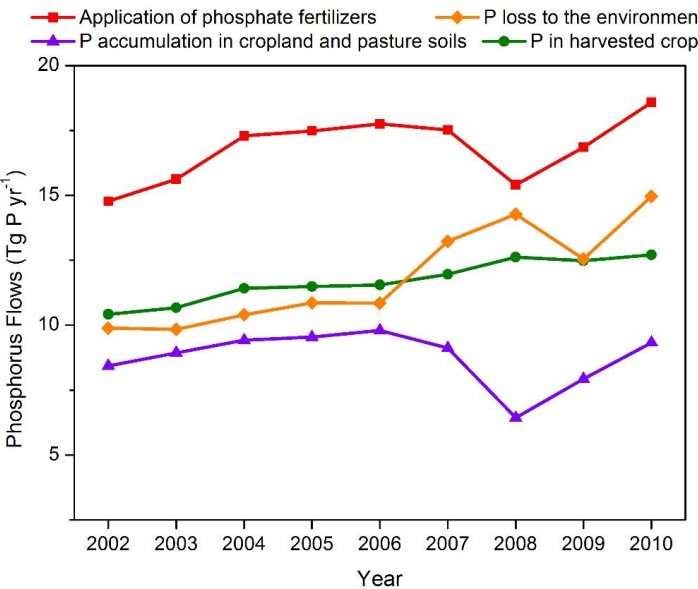


Figure 3: Time series in the four largest global annual P flows, within, in and out of the
agriculture system from 2002 to 2010.

### 3.1.3 Global P fluxes in cropland

Cropland received both the largest fraction (82%) of phosphate fertilizer and 29%

of the manure produced by livestock, as well as all of the recycled human sewage sludge
(Fig. 2). Atmospheric deposition contributed an additional 0.6 Tg P $yr^{-1}$ of inputs to
cropland. Harvesting of cropland removed 11.7 Tg P $yr^{-1}$, which can be divided into
crop products used for human nutrition (9.3 Tg P $yr^{-1}$, including 5.3 for food, 2.7 for
processing, 0.4 for waste and 0.9 for other use) and for livestock feed (2.1 Tg P $yr^{-1}$),
with a small pool in seeds returned to the cropland (0.3 Tg P $yr^{-1}$). On average, 50% of
the P contained in crop residues was recycled to cropland during the study period, with



0.2 Tg P yr⁻¹ lost to the atmosphere from burning of crop residues. The remaining 3.6
Tg P yr⁻¹ contained in harvested crop residues is removed from cropland and
redistributed to livestock and humans. Globally, 3.7 Tg P yr⁻¹ was lost from cropland
soils through leaching and runoff. The sum of all these fluxes results in an annual soil
P accumulation of 6.6 Tg P yr⁻¹ (Fig. 2).
The global cropland PUE averaged 0.46, with a maximum of 0.51 in 2008 and a
minimum of 0.44 in 2006. The annual cropland P accumulation ratio (cropland soil P
accumulation / total P input to cropland) was 23%, which is lower than the
accumulation ratio of 48% found for the overall agriculture system. In countries where
labile P inputs were lower than P removal in crops, the soil's labile P pool was depleted
by 1.9 Tg P yr⁻¹ by harvesting of crop biomass. In countries where labile P inputs are
higher than P removal by crops, the accumulation of soil labile P was 6.0 Tg P yr⁻¹.
Thus, there is an asymmetry between these two groups of countries, with accumulation
being larger than depletion at a global scale. In addition, the global stable P pool in
cropland increased by an average of 5.6 Tg P yr⁻¹ from 2002 to 2010.
**3.1.4 Global P fluxes in pasture**
Figure 2 shows that most P inputs to pasture were from livestock manure (12.7 Tg
P yr⁻¹), with small additional contributions from atmospheric deposition (0.8 Tg P yr⁻¹)
and phosphate fertilizers (0.4 Tg P yr⁻¹). The primary production of pasture incorporates
10.0 Tg P yr⁻¹ of P into grass biomass that is digested by animals, and the leaching and
runoff loss averages 1.7 Tg P yr⁻¹. From all these fluxes, we estimated a global pasture
PUE of 0.72, and a net accumulation of 2.2 Tg P yr⁻¹ in the soil. In the countries where
grass P removal exceeded the labile P inputs, the labile soil P pool was depleted by 1.4
Tg P yr⁻¹. In the countries where the labile P input exceeded grass P removal, an average
of 5.3 Tg P yr⁻¹ was transferred from the labile to the stable soil P pool from 2002 to



2010.

### 3.1.5 Global P fluxes in livestock

The annual P input to livestock was 25.6 Tg P yr$^{-1}$, with most of contributions from
grazed grass (10.0 Tg P yr$^{-1}$) and processed feed (10.0 Tg P yr$^{-1}$). The economic P output
in the form of livestock products averaged 1.5 Tg P yr$^{-1}$, which gives a PUE of 0.06.
Averages of 29% and 56% of the P produced in livestock manure were recycled into
cropland and pasture, respectively; the rest of this manure (4.3 Tg P yr$^{-1}$) was lost to the
environment.

### 3.1.6 Global P fluxes in human use

Humans receive an annual input of 14.0 Tg P yr$^{-1}$ from harvested crop products,
livestock products, and the use of detergents and other products manufactured from
phosphate rock. Although P inputs as food (crop food and livestock products) amounted
to 6.8 Tg P yr$^{-1}$, humans only absorbed 3.0 Tg P yr$^{-1}$ (44%), the remainder being either
wasted before consumption (e.g., in food processing) or transferred back to livestock
as processed feed. Thus, only 14.3% of the total P inputs into the agriculture system
end up as food being actually consumed by humans. P lost to the environment by human
use amounts to 2.6 Tg P yr$^{-1}$, which is divided among 2.2 Tg P yr$^{-1}$ lost through
inefficient processing and excreta and 0.4 Tg P yr$^{-1}$ through bioenergy-related
emissions. The fate of non-recycled P in human waste was not separated between
bodies of water (untreated sewage) and landfill.

### 3.2 Regional P budgets

Cropland and pasture soils accumulated 59.6 and 19.4 Tg P from 2002 to 2010,
respectively. For croplands, the net P accumulation in the stable P pools amounted to
52.7 Tg P, and the remaining 6.9 Tg P accumulated in soil labile pools. For pasture, the
accumulation in the stable P pool was 25.0 Tg P, but 5.6 Tg P was transferred from the



stable P pool to be incorporated by grass in regions where P inputs are lower than grass
P uptake.

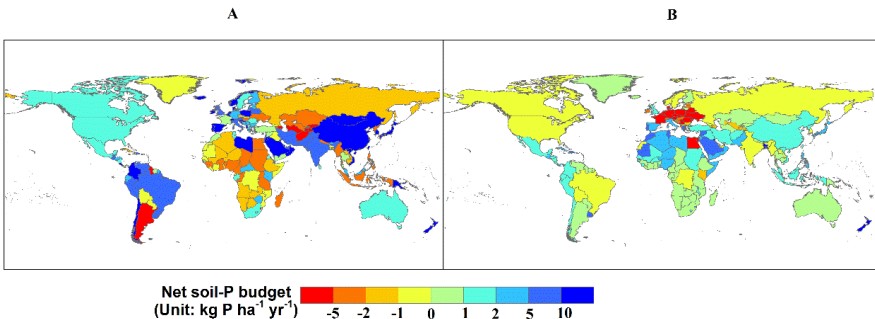

Figure 4: Map of global net soil P budgets (positive values, increase; negative values,
decrease) for (A) cropland and (B) pasture.
Those global numbers mask large regional differences (Table 1, Fig. 4A). About
32% of the global cropland area (in 75 countries) had annual soil P deficits from 2002
to 2007. This fraction increased to 50% in 2008 and 2009, at the time of the global
financial crisis, as a result of high P prices and the resulting reduction in fertilizer
application (Cordell *et al*. 2009, 2012), but returned close to the decadal mean value in
2010. On average, 48% of cropland P uptake was supplied by stable P that accumulated
in previous years according to the equations in section 3 of the Supporting Information.
Including the United States, France, Russia, Argentina, and Paraguay, 89 countries had
labile P inputs into cropland that were lower than crop P removal from 2002 to 2010.
However, if we consider stable P inputs, cropland soil still presented a net soil P surplus
in the United States during the same period. For pasture, a slightly smaller proportion
of the total global pasture area (43%) had a net annual soil P deficit from 2002 to 2010,
mostly in Europe and North America. However, only 48 countries had labile P inputs
into pasture that were lower than the P removal in grass.





### 3.2.1 Regional cropland budgets


Examining Figure 4A reveals that cropland in all African countries experienced an
annual soil P deficit, especially in western and central Africa, with soil P loss rates per
unit area ranging from 2.5 kg P ha$^{-1}$ yr$^{-1}$ in 2002 to 2.7 kg P ha$^{-1}$ yr$^{-1}$ in 2010. In contrast,
cropland in Eastern Asia accumulated 23.4 kg P ha$^{-1}$ yr$^{-1}$ during the period from 2002
to 2010, a cumulative storage equivalent to more than four years of P fertilizer
application. Cropland in Oceania, Europe, and the Caribbean and Central America also
annually accumulated P in their soils. Cropland soils in North America and South
America accumulated P from 2002 to 2007, but experienced temporary P deficits from
2008 to 2010. Yet despite this, crop yields did not decrease from 2008 to 2010 in those
two regions, probably because of the re-mobilization of P that accumulated in stable
pools. Cropland soils in western and central Asia were nearly balanced, with a mean
areal flux of 0.2 kg P ha$^{-1}$ yr$^{-1}$.
Considering the different countries (Fig. 4A), the largest cumulative soil P increase
was found in China (34.6 Tg P) for the 9 years from 2002 to 2010, followed by India
(11.4 Tg P) and Brazil (3.6 Tg P). Pakistan (1.8 Tg P), the United States (1.8 Tg P),
and New Zealand (1.8 Tg P) also had net soil P accumulation, yet of a smaller
magnitude. These six countries accounted for 77% of the global accumulation of P in
countries where cropland had a positive soil P balance. Furthermore, a large amount of
P accumulated in the soil labile P pools of cropland in China and India, at about 20.0
and 4.5 Tg P, respectively; however, in the United States, about 6.0 Tg P accumulated
in the cropland stable P pool from 2002 to 2010; thus, 4.2 Tg P was absorbed from the
previous cropland soil P. In contrast, most African countries experienced persistent
cropland soil P deficits from 2002 to 2010. This was especially true in Nigeria, which
had a cumulative deficit of 1.7 Tg P (Fig. 4A). We also found cumulative soil P deficits



in Russia, the Ukraine, and Kazakhstan, but with a smaller magnitude (1.1, 0.9, and 0.7
Tg P, respectively) for the 9 years. Comparing the rates of change of crop soil P per
unit area, New Zealand had the fastest rate of increase (>100 kg P ha$^{-1}$ yr$^{-1}$), whereas
Argentina had the fastest rate of decrease (–7.9 kg P ha$^{-1}$ yr$^{-1}$). In terms of the difference
between inputs and outputs, loss rates in Argentina were about five times input rates.
**3.2.2 Regional pasture budgets**
We found mainly net losses of P in pasture soils (Fig. 4B), most likely because of
the net removal of P through animal grazing followed by the export of manure P to
enrich cropland soils. Pasture soil P loss rates per unit area in Europe averaged 0.4 kg
P ha$^{-1}$ yr$^{-1}$ and reached high values in countries (Denmark, Luxembourg, Germany, and
Belgium) with intensive livestock production systems (Chang *et al.*, 2015) and large
grass consumption by livestock, with loss rates >10 kg P ha$^{-1}$ yr$^{-1}$). North American
pastures had a smaller average loss rate of about 0.1 kg P ha$^{-1}$ yr$^{-1}$. The United States,
India, and Russia had the largest cumulative P deficits, at 2.1, 1.5, and 0.7 Tg P,
respectively, from 2002 to 2010. In contrast, pasture in the Caribbean and Central
America had greater P inputs than P removals. Consequently, these regions had the
largest soil P accumulation rates. Pasture in Northern and Eastern Africa also had net
soil P accumulation. For instance, Mauritania, Tunisia, and Morocco had net soil P
accumulation rates of 9.8, 9.4, and 5.5 kg P ha$^{-1}$ yr$^{-1}$, respectively. The reason for this
excess is not clear, but one possibility is that these countries apply P fertilizer to some
of their pasture.



Table 1: Regional annual agricultural P budgets and P-use efficiency (PUE)

| Subsystem | World | Eastern and Southern Africa | Northern Africa | Western and Central Africa | Eastern Asia | Southern and Southeastern Asia | Western and Central Asia | Oceania | Europe | North America | Caribbean and Central America | South America |
|---|---|---|---|---|---|---|---|---|---|---|---|---|
| Agricultural land P budget (Tg P yr⁻¹)[1] | | | | | | | | | | | | |
| Cropland | 6.6 | -0.1 | -0.1 | -0.2 | 4.1 | 1.4 | 0.0 | 0.3 | 0.7 | 0.3 | 0.0 | 0.3 |
| Pasture | 2.2 | 0.1 | 0.5 | 0.3 | 0.5 | 0.2 | 0.2 | 0.4 | -0.3 | -0.1 | 0.1 | 0.3 |
| Agricultural land P budget per unit area (kg P ha⁻¹ yr⁻¹)[1] | | | | | | | | | | | | |
| Cropland | 4.7 | -1.0 | -1.5 | -2.7 | 23.4 | 4.1 | 0.2 | 5.2 | 2.8 | 1.5 | 3.8 | 2.3 |
| Pasture | 0.4 | 0.2 | 1.6 | 0.9 | 1.0 | 0.4 | 0.6 | 0.8 | -0.4 | -0.1 | 3.4 | 0.4 |
| Food consumption (Tg P yr⁻¹) | | | | | | | | | | | | |
| Crops | 5.3 | 0.2 | 0.3 | 0.2 | 1.4 | 1.3 | 0.3 | 0.0 | 0.8 | 0.4 | 0.0 | 0.3 |
| Meat | 0.71 | 0.01 | 0.01 | 0.01 | 0.29 | 0.05 | 0.01 | 0.01 | 0.17 | 0.10 | 0.00 | 0.05 |
| Eggs | 0.16 | 0.00 | 0.00 | 0.00 | 0.07 | 0.02 | 0.01 | 0.00 | 0.03 | 0.02 | 0.00 | 0.01 |
| Milk | 0.60 | 0.01 | 0.03 | 0.00 | 0.04 | 0.13 | 0.04 | 0.01 | 0.18 | 0.09 | 0.01 | 0.05 |
| PUE | | | | | | | | | | | | |
| Cropland | 0.46 | 0.80 | 0.84 | 1.51 | 0.27 | 0.43 | 0.64 | 0.31 | 0.54 | 0.57 | 0.53 | 0.63 |
| Pasture | 0.72 | 0.77 | 0.61 | 0.46 | 0.58 | 0.80 | 0.61 | 0.42 | 1.25 | 0.98 | 0.37 | 0.75 |
| Livestock | 0.06 | 0.02 | 0.02 | 0.01 | 0.08 | 0.05 | 0.04 | 0.04 | 0.09 | 0.08 | 0.03 | 0.03 |
| Food | 0.45 | 0.60 | 0.50 | 0.64 | 0.40 | 0.64 | 0.44 | 0.26 | 0.28 | 0.32 | 0.68 | 0.42 |
| International trade of P in commodities (Tg P yr⁻¹)[2] | | | | | | | | | | | | |
| Crops | - | 0.02 | 0.12 | 0.03 | 0.35 | 0.03 | 0.12 | -0.07 | -0.02 | -0.41 | 0.03 | -0.24 |
| Meat | - | 0.000 | 0.001 | 0.000 | 0.013 | 0.000 | 0.002 | -0.004 | -0.002 | -0.005 | 0.001 | -0.010 |
| Eggs | - | 0.0001 | 0.0000 | 0.0000 | 0.0001 | -0.0004 | 0.0001 | 0.0000 | 0.0000 | -0.0002 | 0.0000 | -0.0001 |
| Milk | - | 0.000 | 0.003 | 0.001 | 0.004 | 0.005 | 0.004 | -0.015 | -0.012 | 0.003 | 0.001 | -0.001 |
| Fertilizer | - | 0.05 | -0.67 | 0.03 | 0.00 | 1.45 | -0.05 | 0.21 | -0.37 | -1.00 | 0.06 | 1.10 |

[1] The positive values represent a soil P surplus, whereas negative values represent a soil P deficit.

[2] The positive values represent net P importers, whereas negative values represent net P exporters.



### 3.3 Phosphorus-use efficiencies in different regions

Table 1 gives the values of PUE for cropland, pasture, livestock, and food (human use) in the world's different regions. Globally, 116 countries have cropland PUE values above the global mean value of 0.46, mostly in Africa, and these countries account for 64% of the global cropland area. In addition, 16% of the countries had a PUE of around 0.6 (0.55 to 0.65). In particular, African countries had the highest overall cropland PUE (≥0.80) because of their low P input. On the other hand, Eastern Asia and Oceania have cropland PUE below the global average. Conversely, pasture had high PUE in Europe (1.25) and North America (0.98) but low values in Africa (≤0.77) and particularly low values in the Caribbean and Central America (0.37). P removal from pasture exceeded P inputs in Europe, resulting in pasture PUE > 1, largely because of P inputs from feed given to animals.

The livestock subsystem generally had a low PUE (<0.1), with the highest values in Europe, North America, and Eastern Asia (Table 1). Regarding human food PUE, our data indicate that only 25% to 40% of the P in food products in Eastern Asia, Oceania, Europe, and North America is actually consumed by humans (Table 1). The resulting low PUE of human use in these regions results from both large P inputs and high food waste. Eastern and Southern Africa, Western and Central Africa, Southern and Southeastern Asia, and the Caribbean and Central America had the highest PUE for human use, with more than 60% of P in food being consumed by humans. Globally, most of the P consumed by humans (78%) originates from crops, and the fraction of P from livestock differs among regions; it ranges from 35% of the total human food P consumption in Oceania, Europe, and North America to 10% in less developed regions (Africa and the Caribbean and Central America) and to 4% in Western and Central Africa.





**3.4 P Flows through international trade**

Approximately 2.1 Tg P yr$^{-1}$ entered into international trade in 2010, amounting to about 17% of the total harvested crop P (Figure 5). The remainder (10.6 Tg P yr$^{-1}$) is consumed domestically. Differences in crop types as a result of their specific P content (Table SI-1) strongly determine the magnitude of the traded P fluxes. For example, 37% of the P in soybean and 27% of the P in wheat produced each year were traded internationally in 2010. Also significant fractions of the P in maize, other cereals, and fruit were traded internationally, but almost all of the P in sugar crops and fiber were consumed or processed in the countries where they were grown.

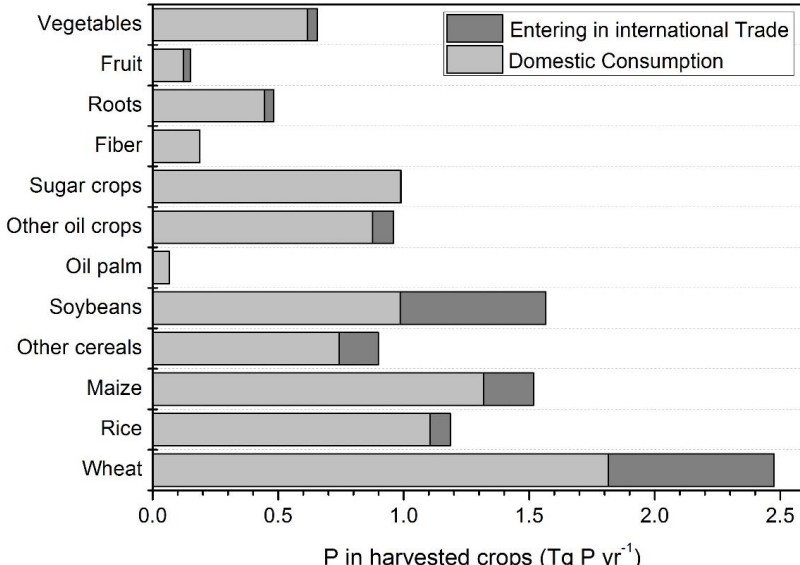

Figure 5: P flows embedded in different crop products, including the fraction of these flows entering into international trade circuits vs. being used for domestic consumption for the year 2010.



Considering the P fluxes in phosphate fertilizers and food products, we examined
how international trade influences regional P budgets and redistributes P between
regions. We found that Southern and Southeastern Asia have the largest net P imports
(Table 1), with imports of phosphate fertilizer amounting to 1.4 Tg P yr$^{-1}$ and P exports
as food products being much smaller, mainly to China and South Korea. South America
is the second-largest exporter of P in food, but imports 56% of its P fertilizer. North
America is a large exporter of P in both crop products and fertilizer, yet it also imports
P-rich milk products. Most European countries imported nearly all their phosphate
fertilizers, but Europe as a whole is a net exporter because of large exports (0.9 Tg P
yr$^{-1}$) from Russia (Figure 6). Western European countries were the main exporters of
P-rich livestock products. Some Northern African countries (especially Morocco and
Tunisia, which have the largest mines of P-rich ores), exported a total of 0.7 Tg P yr$^{-1}$
in fertilizer. The remaining regions (Eastern and Southern Africa, Northern Africa, and
the Caribbean and Central America) imported P in both food and fertilizer, although
much less than other regions (Table 1).
Figure 6 illustrates the disparities among countries with respect to the role of
international trade in crops, livestock, and fertilizer for the main exporters and
importers. Based on data for all 224 countries, a country can be categorized into one of
the following four groups (Figure 7):

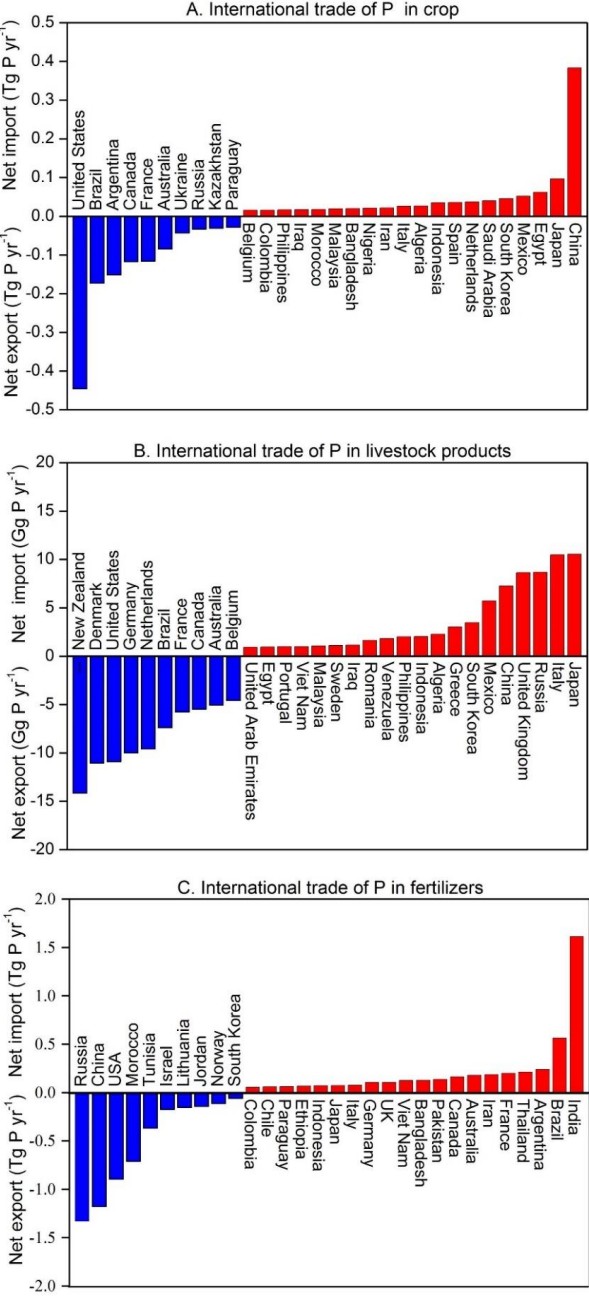


Figure 6: Annual P flows embedded in traded crop products (A), livestock products (B),

and fertilizers (C) in 2010. By convention, a positive flow is P received (imported) by

a country.

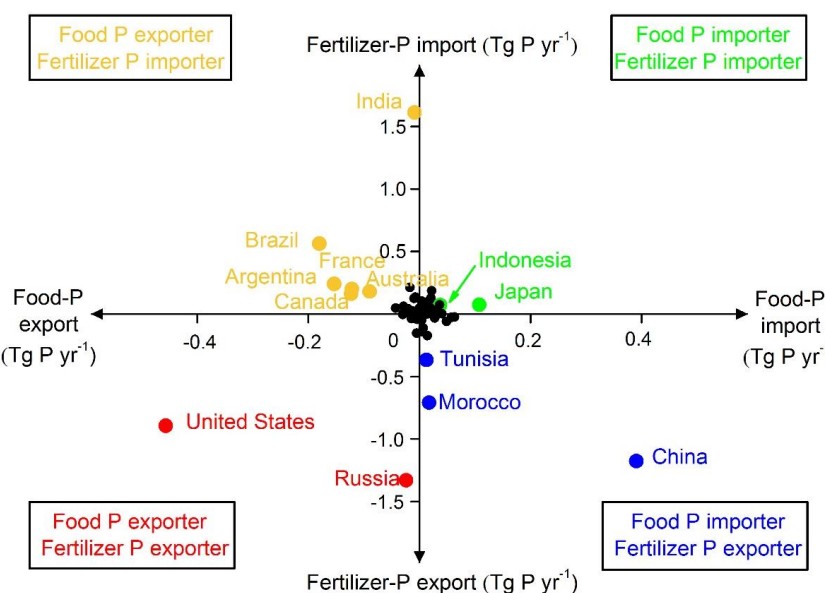


Figure 7: Groupings of the countries based on whether they import or export P through their international trade in food and fertilizer.



Food and fertilizer P exporters: P storage in these countries has been decreasing due to their international exports of both fertilizer and food. Examples include the United States and Russia.




Food P importers and fertilizer P exporters: This group mainly comprises countries that export phosphate fertilizers and import food to meet domestic consumption. Examples includes Tunisia, Morocco, and China.




Food P exporters and fertilizer P importers: These countries have high food and livestock production, but this depends strongly on phosphate fertilizer imported from other countries. Examples include Brazil, Argentina, Canada, France, Australia, and India.





Food and fertilizer P importers: These countries depend on imports for both food and fertilizers; they are thus vulnerable to economic shocks that result from changing food prices. Examples include Japan and Indonesia.






International trade affects the global P cycle by physically moving the P contained
in traded crops, livestock products, and phosphate fertilizers (Grote *et al.*, 2005).
Imports of P fertilizers accounted for 55% and 79%, respectively, of the total
application of P fertilizer for countries that are food P exporters and fertilizer P
importers or food and fertilizer P importers. The P trade in food followed a similar trend.
Countries that are food P importers and fertilizer P exporters or food and fertilizer P
importers depended more on food imports than countries that are food and fertilizer P
exporters or food P exporters and fertilizer P importers. International trade also
increased the connections among countries (Table 2). For example, although the United
States and China are clearly major P fertilizer exporters, they also import fertilizer from
each other; 2.6% of the P fertilizer applied in the United States originated in China, and
3.6% of the phosphate fertilizer applied in China originated in the United States. In
addition, 11.4% of the phosphate fertilizer consumption in the United States originated
from Russia, Morocco, Tunisia, and other countries. About 1.5% of Chinese domestic
P consumption originates from the United States, which is higher than the fraction of
domestic P consumption in the United States from China. Countries with no or small
reserves of P-containing minerals imported large amounts of phosphate fertilizer; for
example, imports accounted for 61 and 46% of total P consumed in France and Brazil
(food P exporters and fertilizer P importers), and 76% of total P consumed in Japan.





Table 2: Proportions of total consumption and total international trade accounted for
by P in fertilizer and food imports and exports.

| Group | Proportion (%) | | | |
|---|---|---|---|---|
| | P fertilizer imports as a proportion of total consumption | P fertilizer exports as a proportion of the total international P fertilizer trade | P in food imports as a proportion of total consumption | P in food exports as a proportion of the total international P in the food trade |
| Group Level | | | | |
| Food and fertilizer exporter | 22 | 43 | 7 | 31 |
| Food importer and fertilizer exporter | 5 | 48 | 22 | 5 |
| Food exporter and fertilizer importer | 55 | 5 | 5 | 48 |
| Food and fertilizer importer | 79 | 4 | 28 | 15 |
| Country level | | | | |
| United States (food and fertilizer exporter) | 13 | 18 | 6 | 26 |
| China (food importer and fertilizer exporter) | 2 | 20 | 14 | 2 |
| France (food exporter and fertilizer importer) | 52 | 0 | 19 | 8 |
| Brazil (food exporter and fertilizer importer) | 44 | 1 | 4 | 10 |
| Japan (food and fertilizer importer) | 40 | 0 | 60 | 0 |


**3.5 Uncertainties in soil P changes result from uncertain P concentrations**
We estimated the net cropland soil P balance in 2000 by means of Monte Carlo
simulations, as described in section 2.7. We found a net accumulation of $5.8 \pm 0.6$ Tg
P yr$^{-1}$. More detailed calculations suggest that uncertainty in the crop P concentrations
contributed $\pm 0.2$ Tg P yr$^{-1}$ of the uncertainty in the net cropland soil P balance; this is
because of dominance of the calculations by cereals, which have low uncertainty due





to the narrow range of reported P concentrations (Antikainen *et al*., 2005; COMIFER,
2007; USDA-NRCS, 2009; Waller, 2010). Uncertainty in P concentrations in crop
residues contributed an additional $\pm 0.2$ Tg P yr$^{-1}$ to the total uncertainty, and uncertainty
in P concentrations in the livestock manure applied to cropland added $\pm 0.4$ Tg P yr$^{-1}$.
In addition, the uncertainty in the pasture soil P balance attributed to uncertainty in the
P concentrations in grass biomass and manure was $\pm 1.3$ Tg P yr$^{-1}$. This relative
uncertainty is higher than that for the cropland soil P balance, and this results from the
large range of grass P concentrations found in our review of the available data. See
Table SI-5 for more details.
**4. Discussion**
**4.1 Cropland PUE and P in harvested crops as a function of cropland P inputs**
Figure 8A shows the relationship between the cropland PUE and cropland P inputs
for 35 countries that are large crop producers. PUE decreased exponentially with
increasing input; that is, P was used most efficiently at low application rates. PUE
decreased rapidly as P inputs increased to 10 kg P ha$^{-1}$ yr$^{-1}$, and then decreased more
slowly. High PUE values were associated with countries that had a low P input and a
soil P deficit. This suggests that there is a trade-off between efficient use of P in
cropland and the avoidance of soil P deficits that limit crop yields (Obersteiner *et al*.,
2013). Figure 8A also indicates that cropland soils have a net soil P deficit if their inputs
are lower than 10 kg P ha$^{-1}$ yr$^{-1}$, which is a threshold value that corresponds to PUE =
0.67. Argentina, South Africa, Indonesia, Mexico, and Paraguay are below this
threshold (Figure 9).



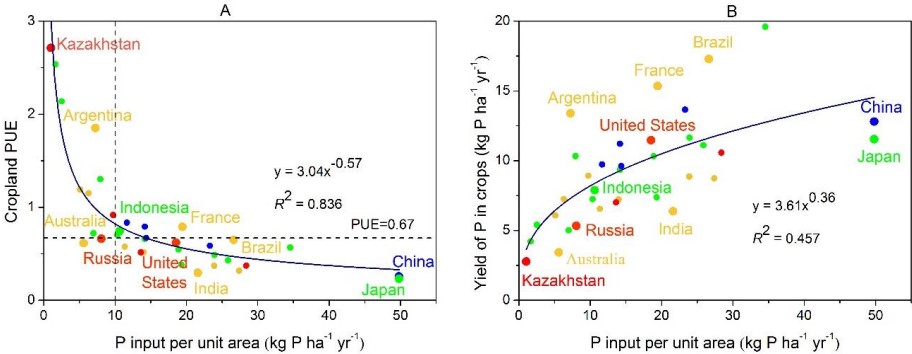


Figure 8: The relationships between P input per unit area of cropland and (A)
phosphorus-use efficiency (PUE) The horizontal line at PUE = 0.67 represents the
global average. (B) P in harvested crops for the 35 largest crop producers representing
90% of global crop. The equations give the fit to the data represented by black curves.

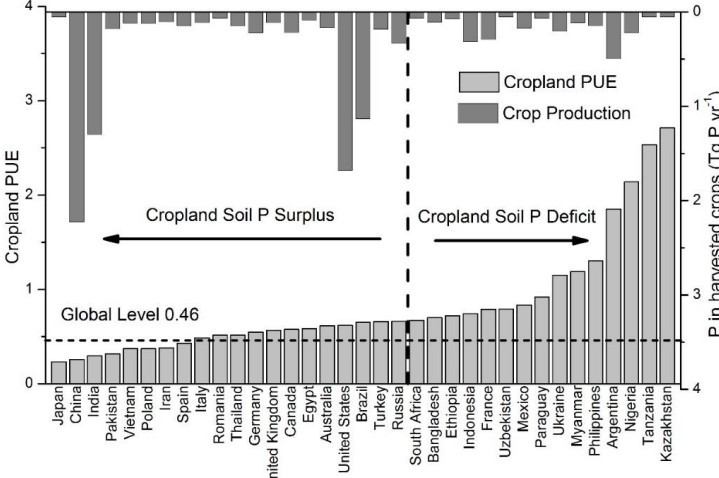


Figure 9: Phosphorus-use efficiency (PUE) and P in harvested crops for the 35 large
countries shown in Fig 8. Cropland soil P surplus or deficit is separated by the vertical
dashed line
P in harvested crops increased exponentially with increasing P inputs, but the
response slowed at high P inputs (Fig. 8B). The P in harvested crops in countries with



cropland PUE > 0.67 (except Argentina) is only half of that in countries with high P in
the harvested crops, such as the United States and China. P in the harvested crops was
very low in Australia due to low cropland P input, which was less than 25% of the
inputs in the United States and China. P already present in the soil may be sufficient to
sustain high crop yields for some time without additional inputs in some countries (e.g.,
France) that formerly had large P fertilization rates, despite currently having a negative
annual P balance. Comparing Figures 8A and 8B suggests that total cropland P inputs
of 20 to 25 kg P ha$^{-1}$ yr$^{-1}$ may be a good compromise that will achieve high yields while
creating a near-equilibrium soil P balance. Both excessive P inputs (e.g., China and
Japan) and low PUE (e.g., India) can lead to high P accumulation in cropland soil,
leading to high losses into the environment.

The data in Figure 8 indicate that different countries face different challenges for

P resource management, implying a need for country-specific policy options and
solutions. Countries like Kazakhstan and Argentina may have to increase P inputs to
their cropland in order to prevent long-term depletion of soil P, which could be realized
by increasing the application of phosphate fertilizer or reducing losses to leaching and
erosion. Countries like France that are currently experiencing a net negative soil P
balance (Fig. 9) following a period of sustained accumulation (Senthilkumar et al. 2012;
van Dijk *et al.* 2016) may need to progressively adjust fertilizer inputs in coming years
to balance inputs with removals and avoid the risk of a long-term soil fertility decline
due to inadequate levels of P. In contrast, countries such as Japan and China are rapidly
accumulating P in cropland soils due high and sustained P inputs, and will urgently
need to consider how to improve their cropland PUE. This could be initiated by
identifying crop types that are being over-fertilized and regions with excessive
application of phosphate fertilizer; they can then consider a range of options such as



precision agriculture (i.e., applying only as much P as the crop requires). We estimate
that if Chinese cropland PUE could be increased to the global average of 0.46 (Fig. 9),
China would save 3.8 Tg P yr$^{-1}$ of phosphate fertilizer, which is equivalent to 60% of
its phosphate fertilizer consumption in 2010. Last, in countries like India where crop P
harvests are lower than average despite high average P inputs and positive soil $\Delta P$,
improvements in agricultural management (such as the use of precision fertilization)
appear necessary. We did not have access to sub-national data for this study, but it is
likely that in a country as large as India, some regions, crop types, or region–crop type
combinations may have excessive or insufficient P input.
**4.2 Pasture P budget, livestock consumption, and international trade**
Figure 10 shows that the soil P balance is negatively related to the flux of P in
livestock products per unit area of pasture. Several western European countries
(Germany, the Netherlands, Denmark, and Belgium) achieve high P yields in livestock
products (defined by the amount of P in livestock products per unit area of pasture),
and all of these countries export livestock products. In these countries, only a small
fraction of livestock manure is recycled to pasture, so there is currently a soil P deficit;
in the long term, this may result in a loss of soil fertility. Therefore, these countries
should increase P fertilization in pasture or import forage or feed to supply the P
required to sustain high livestock production. New Zealand, Australia, and Canada are
also large exporters of P in livestock products. However, given their low-input
production systems and large areas of pasture (Fig. 4B), P removals per unit area
through grazing are much lower than in Western Europe, and the soil P balance of
pasture ranges from slightly negative to slightly positive.



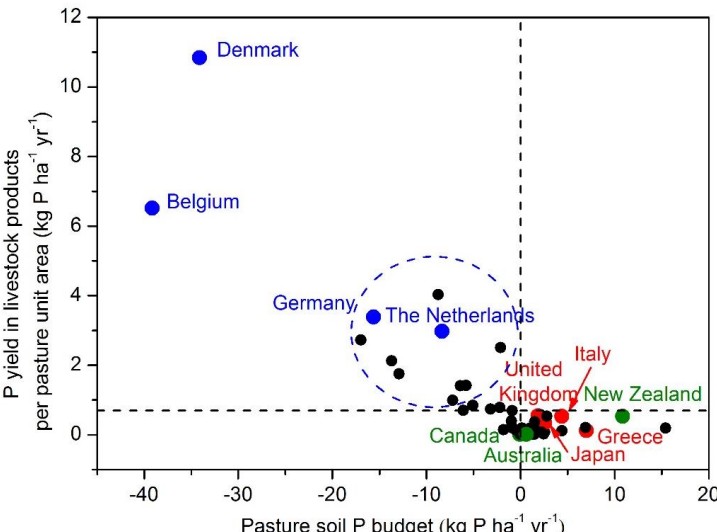


Figure 10: The relationship between the P yield of livestock products, defined by the

amount of P in livestock products per unit area of pasture and the P balance of pasture

soils.

**4.3 Livestock and human food PUE, and trends in P consumption**

Increasing consumption of livestock products by humans is an essential factor that

is responsible for increasing P mining and increasing P inputs to agricultural systems

(Metson *et al.*, 2012; van Dijk *et al.*, 2016). Where socioeconomic development is

improving the income of residents, especially in Africa and the Caribbean and Central

America region, residents are consuming more P from livestock products (Fig. SI-3).

Unfortunately, the livestock PUE in countries in these two regions is much smaller

(0.01 to 0.03) than the global average of 0.06 (Table 1), indicating that only a small

proportion of livestock P inputs is used by humans. This may be because countries in

these regions are primarily importers of livestock products. Therefore, animal

husbandry has important implications for global P security and special attention will be

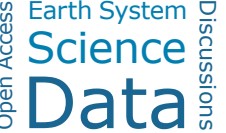

required to improve livestock PUE (Wu *et al.*, 2014). If livestock PUE reaches the
global level of 0.06 in these two regions, both regions could more than double their
livestock production, by about 0.16 Tg P yr$^{-1}$.
In addition, the management of manure differs greatly among regions due to their
different livestock production systems. The yield of livestock products is very low in
African countries, resulting in low livestock PUE. Almost all livestock manure is
applied to cropland, where this resource is an important P input. In contrast, with
application of phosphate fertilizer to pasture in Europe and Eastern Asia, only a small
fraction of livestock manure is recycled for pasture (36 and 17%, respectively); a larger
fraction of the manure is applied to cropland in Eastern Asia (40%) and Europe (60%).
Consequently, improving the manure utilization efficiency and applying more livestock
manure to pasture will be important strategies in Eastern Asia and Europe (Wu *et al.*,

2014).

As shown in Section 3.3, only 45% of the P that enters the food production
subsystem was absorbed by humans; thus, large amounts of food (and the P it contains)
are wasted, although some parts of the wastes were consumed by livestock. Despite this
recycling, 2.2 Tg P yr$^{-1}$ flowed into the environment as wastes either before or after
food consumption, and only 14.3% of the total P inputs to the agriculture system ended
up in food consumed by humans. In Eastern Asia, Oceania, Europe, and North America,
the PUE of human food was very low, reflecting the high proportion of livestock
products in the diet and a high degree of waste. Therefore, decreasing food waste before
consumption, recycling P in food waste, and better treatment of organic wastes could
significantly decrease the amount of P required to support humans (Metson et al., 2012;
van Dijk *et al.*, 2016). In Eastern Asia, Oceania, Europe, and North America, fully
absorbing the 45% of the P that enters food produced for humans could reduce





agricultural inputs of P by 0.7 Tg P yr$^{-1}$ globally. Thus, decreasing food waste and
improving the PUE of human food represent key challenges that must be solved to
achieve sustainable P management.
Population increases and dietary changes are requiring higher P inputs in cultivated
land and increased mining of P ores (Grote *et al*., 2005; Foley *et al*., 2011). From 2002
to 2010, this mining increased by 33% in our estimate, during a period when the global
population and per capita food P consumption increased by 10 and 5%, respectively. In
2010, humans consumed 8.0% and 3.8% more P in livestock products and crops,
respectively. Since livestock PUE was much lower than cropland PUE, consumption of
more livestock products resulted in lower external P inputs in food that flowed into the
human subsystem; this proportion decreased from 36% in 2002 to 31% in 2010.
Therefore, consuming more livestock products will require increasing P inputs. Thus,
human dietary shifts may have been responsible for half of the increase of P ore mining.
**4.4 International trade and global P flows**
International trade also increased the connections among countries. Whether
international trade is good or bad for humans and the environment in terms of its impact
on the management of P resources is a complex question. International trade can
increase cropland P deficits if countries that export large amounts of P in crop and
livestock products do not counteract these exports by increasing inputs of phosphate
fertilizer to soils. For example, Argentina exported lots of food to other countries (about
0.15 Tg P yr$^{-1}$), and has developed a serious cropland soil P deficit of 0.38 Tg P yr$^{-1}$
(10.3 kg P ha$^{-1}$ yr$^{-1}$). Massive P imports through trade can result in an excess supply of
P to cropland soils as manure (Schipanski and Bennett, 2012), with potentially
significant negative environmental effects. On the one hand, trade can hamper the
proper recycling of P resources from wastes and manure to agricultural soils through



local food webs (Schipanski and Bennett, 2012). On the other hand, trade may
contribute to more efficient use of P resources if traded products flow from countries
with lower PUE to countries with higher PUE, as is generally observed for water
resources (Dalin et al., 2014). This confirms that more integrated studies are required
to fully assess the effects of trade on P resource recycling, efficiency, and conservation.
Our study identified world regions and countries with lower PUE and others with high
PUE, and regions and countries with net loss of P in soils and others with net gain. This
provides valuable information to policymakers on how to improve the trade
relationships for a global optimization of PUE and therefore global food security.
**4.5 Comparison with previous studies**
Previous studies have estimated P flows in agriculture at a global scale (Smil, 2000;
Sheldrick *et al*., 2003; Liu *et al*., 2008; Cordell *et al*., 2009; Bouwman *et al*., 2009,
2013; Potter *et al*., 2010; MacDonald *et al*., 2011). However, to the best of our
knowledge, the present analysis provides the first consistent multi-year overview of the
P flows in agriculture. In addition, it provides national and regional P budgets,
calculates agricultural PUE, and quantifies P fluxes in international trade based on a
combination of datasets for cropland and pasture inputs (fertilizers, manure,
atmospheric deposition, and recycling of crop residues) and outputs (crop harvests,
residue removal, and P loss by burning and leaching or surface runoff into bodies of
water). For data from 2000, our results are consistent with the abovementioned studies
for most P flows (Table 3). For data from 2000, our results are generally consistent with
those in the previous studies for cropland soil P inputs, harvested crop P, cropland soil
P lost by erosion or surface runoff into bodies of water, pasture soil P inputs, and
harvested grass P (Table 3). However, methods, data sources, and system boundaries
differed among the studies, making an accurate comparison difficult. Our estimate of a



net accumulation of 5.8 ± 0.6 Tg P yr$^{-1}$ is in line with the reported net accumulation in
soils, which ranged between 0 and 11.5 Tg P yr$^{-1}$ (Smil, 2000; Bennett *et al.*, 2001;
Bouwman *et al.*, 2009; MacDonald *et al.*, 2011), but disagrees with the estimate of Liu
*et al.* (2008), who calculated a net loss of 9.6 Tg P yr$^{-1}$. The difference from the present
results can be explained by accounting for large P losses (19.3 Tg P yr$^{-1}$) due to soil
erosion caused by land use change and over-grazing.  The quantification of erosional
losses of P from arable land is prone to high uncertainties due to the unknown amount
of redeposited soil material, and other studies have reported much lower losses (e.g.,
2.5 Tg P yr$^{-1}$; Quinton et al., 2010).
Table 3: Comparison of the present results for P flows and budgets in 2000 with results
of other studies at a global level  (Tg P yr$^{-1}$).

|  | Global P flux | Previous studies | Our study | Reasons for differences |
|---|---|---|---|---|
| | Fertilizer input | 14–15 [1–3] | 13.7 | – |
| | Animal manure to cropland | 6–8 [2,3] | 6.7 ± 0.4 | Method |
| | Human sewage sludge to cropland | 1.5 [1,3] | 1.3 | Method |
| | Crop production | 8.2–12.3 [1–5] | 10.2 ± 0.4 | Boundary/Data |
| Cropland | Crops (human food) | 3.5 [3] | 4.8 ± 0.2 | Method/Data |
| | Crops (animal feed) | 2.6 [3] | 1.9 ± 0.1 | Data |
| | Crop residues | 3.75–4.5 [1–2] | 6.7 ± 0.2 | Method/Data |
| | Recycling of residues | 1–2.2 [1–3] | 3.5 ± 0.1 | Method/Data |
| | Leaching and runoff from cropland | 4 [6] | 3.2 | Method |
| | Livestock manure | 17.1–24.3 [5,7,8] | 22.3 ± 1.3 | Method/Data |
| | Manure wasted (released into the environment) | 2–8 [1–3] | 4.1 ± 0.2 | Method/Data |
| Pasture | Grass | 6–12.1 [3,4] | 8.9 ± 1.3 | Method/Data |
| | Animal feed additives | 0.9 [3] | 1.4 | Data |
| | Leaching and runoff from pasture | 1.0 [5] | 1.6 | Method |
| Humans | Excreta | 3–3.3 [1,3] | 2.8 | Method |

Sources: 1. Liu *et al.*, 2008; 2. Smil, 2000; 3. Cordell *et al.*, 2009; 4. MacDonald *et al.*, 2011; 5. Bouwman *et*
*al.*, 2009; 6. Bouwman *et al.*, 2011; 7. Sheldrick *et al.*, 2003; 8. Potter *et al.*, 2010.

The main cropland P fluxes estimated in our study agreed with previous results,
except for the production and recycling of crop residues (Table 3). Smil (2000) and Liu
*et al.* (2008) used harvest index data (defined as the ratio of total aboveground biomass



to crop residues) for estimating the P in crop residues, whereas we estimated P in crop
residues by combining data from Liu *et al*. (2008) and FAO. MacDonald *et al*. (2011)
estimated that 29% of the global cropland area was subject to soil P deficits in 2000,
which is similar to our estimate (32%) based on data from 2002 to 2010. In addition,
our estimate of 22.3 Tg P yr$^{-1}$ in animal manure for the livestock subsystem in 2000 is
within the reported range of 17.1 to 24.3 Tg P yr$^{-1}$ from Potter *et al*. (2010). We defined
global cropland PUE as the ratio of P in harvested crops to total P inputs, without
accounting for recycling of crop residues. Under this definition, global PUE was
estimated to be 0.43 by Liu *et al*. (2008) and 0.40 by Smil (2000), both of which are
comparable to our estimate of 0.46 from 2002 to 2010. Since we applied the same
methods across the globe to calculate agricultural P fluxes, we were able to compare
the P fluxes and budgets for different regions and countries on a consistent basis. This
information is of critical importance for the development of more appropriate
agricultural policy and to support the development of technological and other solutions
for different types of countries, which better integrate cultivated ecosystems, livestock
production, and the human food supply.
**4.6 Limitations and novelty of our study**
Due to limited data sources for some parameters, our study and most previous
studies focused on P in livestock products and manure as the outputs of the livestock
system, and did not consider the fate of P in non-edible livestock products (e.g., bones,
blood, leather products). Xu et al. (2005) pointed out that from 12 to 23% and 72% of
P were contained in livestock meat and bones, respectively. If these percentages are
applied to our data, this gives an annual flux of 2.5 Tg P yr$^{-1}$ in the bones of slaughtered
animals. Although most livestock bones are currently wasted or landfilled, some
countries have begun to use them as fertilizers, protein sources, and condiments (Wu



and Ma, 2005; Li, 2008). In addition, as we focused on the annual P budgets for
livestock and human beings, we did not account for P accumulation in humans. From
2002 to 2010, the global population increased by $635\times10^6$ persons. If we assume that a
typical adult body contains 600 g of P, then about 0.38 Tg more P would have
accumulated in humans. Therefore, the annual human P accumulation would be 0.04
Tg P yr$^{-1}$, accounting for only 0.3% of the P inputs into humans.
Despite the abovementioned limitations in our study, we were able to achieve some
interesting and novel results. First, we have provided a detailed and harmonized
summary of the P fluxes as inputs and outputs for the agricultural system and the
internal P flows within the agricultural system at national, regional, and global scales.
In addition, we have characterized the P budgets and P-use efficiencies in the
subsystems of the overall agricultural system, and have discussed their influences and
impacts. Finally, we have discussed how changes in population, diets, and food
consumption have influenced global mining of P ore and how international trade has
influenced P fluxes. These insights will support the development of policies to use P
more sustainably at national, regional, and global levels.
**Data availability**
The global and regional phosphorus budgets and their PUEs in agricultural systems
is publicly available at https://doi.pangaea.de/10.1594/PANGAEA.875296.
**Conclusion**
The estimation of global and regional phosphorus budgets in agricultural systems,
as well as their PUE, is a major effort by anthropogenic nutrient cycle research
community that requires lots of work. We quantified in detail the P inputs and outputs
of cropland and pasture, and the P fluxes through human and livestock consumers of
agricultural products, at global, regional, and national scales from 2002 to 2010. The



results reveal the significant and imbalanced P budgets in cropland and pasture. The
hot spots of cropland P budgets shifted from increasing P accumulation in Eastern Asia
countries to increasing soil P deficits in African countries, while European and North
American pasture had a soil P deficit. There presents great differences among the values
of PUE for or cropland, pasture, livestock, and food at global, regional, and national
scales. PUE decreased exponentially with increasing input; that is, P was used most
efficiently at low application rates; meanwhile, P in harvested crops increased
exponentially with increasing P inputs, but the response slowed at high P inputs.
International trade played a significant role in P redistribution among countries through
the flows of P in fertilizer and food among countries. It can mitigate regional P
imbalances in agricultural soils, by optimizing phosphate fertilizer application and
recycling P.

**Acknowledgments**
This study was supported by the National Natural Science Foundation of China
(41571022, 41625001), the Beijing Natural Science Foundation (Grant 8151002), the
National Science and Technology Major Project (2015ZX07203-005), and a Synergy
Grant (ERC-2013-SyG-610028 IMBALANCE-P) from the European Research
Council.

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

Notes. Coefficients to convert livestock numbers into manure nitrogen quantities

from national sources. Organisation for Economic Cooperation and Development,

Paris, 1991.

Ott, C., Rechberger, H. The European phosphorus balance. Resour., Conserv. Recy.,

60, 159-172, 2012.

Penuelas, J., Poulter, B., Sardans, J., Ciais, P., Van Der Velde, M., Bopp, L., Boucher,

O., Godderis, Y., Hinsinger, P., Llusia, J., Nardin, E., Vicca, S., Obersteiner, M.,

and Nardin, E. Human-induced nitrogen–phosphorus imbalances alter natural and

managed ecosystems across the globe. Nat. Commun., 4, 2934, 2013.

Potter, P., Ramankutty, N., Bennett, E. M., & Donner, S. D. Characterizing the spatial

patterns of global fertilizer application and manure production. Earth Interact.,

14(2), 1-22, 2010.

Quinton, J. N., Govers, G., Van Oost, K., and Bardgett, R. D. The impact of agricultural

soil erosion on biogeochemical cycling. Nat. Geosci., 3(5), 311-314, 2010.

Ringeval, B., Nowak, B., Nesme, T., Delmas, M., and Pellerin, S. Contribution of

anthropogenic phosphorus to agricultural soil fertility and food production. Global

Biogeochem. Cy., 28(7), 743-756, 2014.

Sattari, S. Z., Bouwman, A. F., Giller, K. E., and van Ittersum, M. K. Residual soil

phosphorus as the missing piece in the global phosphorus crisis puzzle. PNAS,

109(16), 6348-6353, 2012.

Schipanski, M. E., Bennett, E. M. The influence of agricultural trade and livestock

production on the global phosphorus cycle. Ecosystems, 15, 256-268, 2012.

Scholz, R. W., Ulrich, A. E., Eilittä, M., and Roy, A. Sustainable use of phosphorus: a



finite resource. Sci. Total Environ., 461, 799-803, 2013.
Senthilkumar, K., Nesme, T., Mollier, A., and Pellerin, S. Regional-scale phosphorus
flows and budgets within France: the importance of agricultural production
systems. Nutr. Cycl. Agroecosys., 92(2), 145-159, 2012.
Sheldrick, W., Syers, J. K., and Lingard, J. Contribution of livestock excreta to nutrient
balances. Nutr. Cycl. Agroecosys., 66, 119-131, 2003.
Smil, V. Phosphorus in the environment: natural flows and human interferences. Annu.
Rev. Mater. Sci., 25, 53-88, 2000.
Steffen, W., Richardson, K., Rockström, J., Cornell, S. E., Fetzer, I., Bennett, E. M.,
Biggs, R., Carpenter, S. R., de Vries, W., de Wit, C. A., Folke, C., Gerten, D.,
Heinke, J., Mace, G. M., Persson, L. M., Ramanathan, V., Reyers, B., and Sörlin,
S. Planetary boundaries: Guiding human development on a changing planet.
Science, 347(6223), 1259855, 2015.
Suh, S., Yee, S. Phosphorus use-efficiency of agriculture and food system in the US.
Chemosphere, 84, 806-813, 2011.
USDA-NRCS. Crop Nutrient Tool: Nutrient Content of Crops. United States
Department of Agriculture, Natural Resource Conservation Service, Washington,

2009.

van Dijk, K. C., Lesschen, J. P., Oenema, O. Phosphorus flows and balances of the
European Union Member States. Sci. Total Environ., 542, 1078-1093, 2016.
Van Vuuren, D. P., Bouwman, A. F., Beusen, A. H. W. Phosphorus demand for the
1970-2100 period: a scenario analysis of resource depletion. Global Environ.
Chang., 20, 428-439, 2010.
Waller, J. C. Byproducts and unusual feedstuffs. Feedstuffs, 9, 18-22, 2010.
Wang, R., Balkanski, Y., Boucher, O., Ciais, P., Peñuelas, J., and Tao, S. Significant





contribution of combustion-related emissions to the atmospheric phosphorus
budget. Nat. Geosci., 8(1), 48-54, 2015.
Wang, R., Tao, S., Balkanski, Y., Ciais, P., Boucher, O., Liu, J., Piao, S., Shen, H.,
Vuoloc, M. R., Valarie, M., Chen, H., Chen, Y., Cozic, A., Huang, Y., Li, B., Li,
W., Shen, G., Wang, B., and Zhang, Y. Exposure to ambient black carbon derived
from a unique inventory and high-resolution model. PNAS, 111(7), 2459-2463,

2014.

Wu, H., Yuan, Z., Zhang, Y., Gao, L., and Liu, S. Life-cycle phosphorus use efficiency
of the farming system in Anhui Province, Central China. Resour., Conserv. Recy.,

83, 1-14, 2014.

Wu, L., Ma, M. The comprehensive utilization of animals' bone in China. Modern Food
Sci. Tech., 83, 38-46, 2005. (in Chinese).
Xu, J., Liu, X., Wang, F., Zhang, F., Ma, W., and Ma, L. Phosphorus balance and
environmental effect of animal production in China. Acta Ecol. Sinica, 11, 2911-
2918, 2005. (in Chinese).