# Peer review of "Global P budgets in agricultural systems and their implications for phosphorus use efficiency"

_Earth System Science Data, 2017_

## Referee Comment (RC1) · Anonymous Referee #1 · 9 Aug 2017

The authors have done a lot of calculation and the paper presented a lot of good data. However, the paper lack of a very clear narrative storyline. The important results were hidden in a lot of data. The authors also lack deep analysis of the model results. It is very difficult for the audiences to identify the most important conclusions and explore the story behind the data. It will be good if the authors can reorganize the text a little bit and highlight the most important results and conclusions. Some specific comments are below:

1. P4 Line 74-79 "Previous research mainly focused on cropland while P fluxes in pasture and livestock production systems received less attention (McDowell and Condron,

2004)". This sentence is not precise and was supported by a literature in 2004 (more than 10 years ago). Actually there are a lot of studies including pasture and livestock production and the terms "agriculture" and "food production system" also include livestock production system in most cases (please see selected reference blow). With the diet change, more attention is attracted to the livestock production system. The logic of the long sentence needs to be further clarified. Do the authors mean "the differences in methodologies, system boundaries, and data sources hamper the assessment of (or make it difficult to assess) PUE"?

Reference Bai ZH, Ma L, Oenema O, et al. 2013. Nitrogen and phosphorus use efficiencies in dairy production in China. Journal of Environmental Quality, 42:990-1001. Cordell D, Drangert J-O, White S. 2009. The story of phosphorus: Global food security and food for thought. Global Environmental Change, 19:292-305. Ma L, Wang FH, Zhang WF, et al. 2013. Environmental assessment of management options for nutrient flows in the food chain in China. Environmental Science and Technology, 47:7260-7268. Liu Y, Villalba G, Ayres R U, et al. 2008. Global phosphorus flows and environmental impacts from a consumption perspective. Journal of Industrial Ecology, 12(2):229-247.

2. There are some problems in the Figure 1 (P6). The biggest problem is that the Human system should not be part of agricultural system. Detergent is not only for agriculture. There are some industrial uses for P, although the fraction is small. Most sludge is produced from urban life and should be in the agricultural system either.

3. The authors used fertilizer for chemical fertilizer, which is not appropriate and will raise a lot of confusions. The term fertilizer includes all kinds of nutrient used into the soil including organic fertilizer. In P6, figure 1, the fertilizer box should be chemical fertilizer. In P9, Line199-201. The dependency indicator can be the ratio of imported chemical fertilizer to the P in all chemical fertilizer or all fertilizer, which both make sense but have different policy implications.

[Figure]

4. Table SI-2. The authors need to polish the crop categories. There are some inappropriate sorting and overlaps for different types of crops. For example the maize and the silage (maize) have overlaps. Popcorn, to my understanding, is the processed maize, but the authors sort it into other cereal.

———————————————————

---

## Referee Comment (RC2) · Anonymous Referee #2 · 22 Sep 2017

The overall intent of the submission, to provide data of P budgets and trends (2002-2010) across countries and continents, is laudable. It should be pointed out that this is a large data set and it required a lot of work to develop. All cells are populated - there are no gaps or missing information in this dataset which would be expected given that some of the entities are tiny. The data includes estimates of agricultural P budgets and P Use Efficiencies (PUE) based on crops, pastures, deposition, runoff/leaching) and trade in food and fertilizer, and other uses which was mainly soap. These calculations were made for every country and for continental regions and for every year and finally averaged over the period. The data are presented in an Excel spreadsheet supported by 7 tables and 3 graphs included as supplemental information. There is

also supporting text which provides additional information on sources, references and calculations. I did not find many anomalies (no fertilizer for Belgium? and a cell ref. issue in Detailed worksheet)

The data in the worksheets are easy to follow. However, there are no units specified on any of the columns which significantly detracted from ease of use. Also, the sources of the data is not readily at hand, this would be very helpful. For example in Table SI-4 FAO is cited as a source of a great deal of data, but the exact reference is not given or even the year of the publication. Is it FAO 2002 cited in the reference? In fact the vast majority of the data is from FAO. Another sources, for the 'Phosphate acid' is the IFA- but again publication is not given and there is no citation. The other source is an atmospheric model reported in Liu et al 2007 but here the citation is not given on the table although it is in the text.

Some of the parameters, too, are curious. For example, the P:N ratios and P contents in SI-3 have no source and it is curious, for example, that the values for dairy and non-dairy cattle is the same. It is not clear why a range is given for P:N and not for P- surely there is uncertainty also for P even if less than P:N. Further, are the same values used for the entire world. And is the P content based on as is or dry weight?

Table SI-5 is a summary for cropland P inputs and outputs. There are ranges given but it si not clear how these were determined. Also, I do not know what recycled crop residue means. Is it crop residue left in the field? I find the values for deposition (atmospheric) for P unexpectedly high. I realize these values have been published and were derived form a model. It would help if the source of the deposition was quantified and presented as a part of the budget, otherwise it looks like it came from outide the boundaries of the model which does not seem correct. The human sewage value also seems high. I tracked the values to Liu et al 2008, which supposed that 30% of urban sewage and 70% of rural sewage was applied to agricultural land. But When I looked at some western countries the ratio was less than 10% which seems more plausible but where did these estimates come from. In any case it is not transparent. In Table

SI-7 cropland PUE> total PUE; this seems wrong and is in contrast to lines 271-272.

Since the justification for this submission is to provide useful data, which I fully support, I think it is incumbent on the authors to make the data and its sources/ computation as transparent as possible both on the SI and the worksheets. They should clearly explain their uncertainties and how ranges were determined. Certainly there is a vast difference in uncertainties among national entities. Also I could not see where country types are defined, perhaps I missed it. Likewise, what is the difference between crop residues and total crop residues? Where did the labile vs stable (0.8 vs 0.2) coefficients come from?

Overall, this is an interesting data set that will be more useful when it is made more transparent. I look forward to seeing and probably using the data set in the future.

Comments on Abstract.

Globally, half of the total P input (21.3 TgP yr-1) into agricultural systems accumulated in agricultural soils during this period, with the rest lost to bodies of water through complex flows. Comment I could not find this in the document

Global P accumulation in agricultural soil increased from 2002 to 2010, despite decreases in 2008 and 2009, and the P accumulation occurred primarily in cropland. Despite the global increase of soil P, 32% of the world's cropland and 43% of the pasture had soil P deficits.

Comment: I could not find this in the document

European and North American pasture had a soil P deficit because continuous removal of biomass P by grazing exceeded P inputs.

Comment: Manure P deposition on pasture will closely match removal of biomass P by grazing; any deficit from harvesting of animals will be quite small, especially if stocking rates are low.

Based on country-scale budgets and trends we propose policy options to potentially mitigate regional P imbalances in agricultural soils, particularly by optimizing the use of phosphate fertilizer and recycling of waste P.

Comment: self-evident.

The trend of increasing consumption of livestock products will require more P inputs to the agricultural system, implying a low P-use efficiency aggravating the P stocks scarcity in the future.

Comment: While technically this is correct, I don't really see a strong connection with the data set. There is more prospects for recycling animal P than N.

---

## Author Comment (AC1) · 17 Oct 2017

Dear Editors and Reviewers: Thank you for your letter and for the reviewers' comments concerning our manuscript entitled "Global and regional phosphorus budgets in agricultural systems and their implications for phosphorus-use efficiency". Those comments are all valuable and very helpful for revising and improving our paper, as well as the important guiding significance to our researches. We have studied comments carefully and have already made the revision. The revised portions are marked in red in this reply. The main corrections in the paper and the responds to the reviewer's comments are as flowing:

[Figure]

**Reviewer: 1 First of all, thank you so much for your great comments, and I have revised all of them as follows: Comments 1: P4 Line 74-79 "Previous research mainly focused on cropland while P fluxes in pasture and livestock production systems received less attention (McDowell and Condron, 2004)". This sentence is not precise and was supported by a literature in 2004 (morethan 10 years ago). Actually there are a lot of studies including pasture and livestock production and the terms "agriculture" and "food production system" also include livestock production system in most cases (please see selected reference blow). With the diet change, more attention is attracted to the livestock production system. The logic of the long sentence needs to be further clarified. Do the authors mean "the differences in methodologies, system boundaries, and data sources hamper the assessment of (or make it difficult to assess) PUE"? Reply: We have rewritten this part and made it further clarified as follow: "Previous research mainly focused on cropland, and P fluxes in pasture and livestock production systems have also received more and more attention recently, especially due to diet change. The differences in methodologies, system boundaries, and data sources have made it difficult to assess the differences in the phosphorus use efficiencies (PUEs) among agricultural sectors and to extrapolate regional findings to the global scale".**

Comment 2: There are some problems in the Figure 1 (P6). The biggest problem is that the Human system should not be part of agricultural system. Detergent is not only for agriculture. There are some industrial uses for P, although the fraction is small. Most sludge is produced from urban life and should be in the agricultural system either. Reply: Thank you so much for your good suggestions. In our revised manuscript, we have re-drawn the Figure 1 and Figure 2. For these new figures, the "Human system" and "detergent and other uses" are not include in the agricultural system, as you suggested. Besides, we have also updated related parts.

Comment 3: The authors used fertilizer for chemical fertilizer, which is not appropriate and will raise a lot of confusions. The term fertilizer includes all kinds of nutrient used into the soil including organic fertilizer. In P6, figure 1, the fertilizer box should be

chemical fertilizer. In P9, Line199-201. The dependency indicator can be the ratio of imported chemical fertilizer to the P in all chemical fertilizer or all fertilizer, which both make sense but have different policy implications. Reply: According to your great suggestions, we have revised all these parts by "chemical fertilizer" replacing "fertilizer", including figures and texts.

Comments 4: Table SI-2. The authors need to polish the crop categories. There are some inappropriate sorting and overlaps for different types of crops. For example the maize and the silage (maize) have overlaps. Popcorn, to my understanding, is the processed maize, but the authors sort it into other cereal. Reply: Thank you so much for your good comments. In our study, we divide crops into maize, rice, wheat and other cereals. In the FAO statics, they put forage and silage (maize, grasses nes, alfalfa and so on) into other crops, and here we also put them into the type of "other crops". For popcorn, we have checked it, and then deleted. In our future research, we will research on much more detailed categories.

Comments 5: It will be good if the authors can reorganize the text a little bit and high-light the most important results and conclusions. Reply: We have rewritten the conclusion part and highlight the most important results as follows (Line 775-796): The results reveal P from phosphate fertilizers was the largest single input flux into the agriculture system, while one half flowed into waters and 1/3 of them accumulated in soil. Chemical fertilizer inputs, P loss to the environment and P harvested in crops presented significantly increasing trends. Global significant imbalanced P budgets in cropland and pasture mask large regional differences, and there also existed differences between cropland and grassland. Compared with cropland, a slightly larger proportion of the total global pasture area had a net annual soil P deficit. The hot spots of cropland P budgets shifted from increasing P accumulation in Eastern Asia countries to increasing soil P deficits in African countries, while European and North American pasture had significant soil P deficit. There presents great differences among the values of PUE for or cropland, pasture, livestock, and food at global, regional, and national scales, and

livestock subsystem generally had the lowest PUE. PUE decreased exponentially with increasing input; that is, P was used most efficiently at low application rates; meanwhile, P in harvested crops increased exponentially with increasing P inputs, but the response slowed at high P inputs. International trade played a significant role in P redistribution among countries, and near one fifth of total harvested crop P entered into international trade. Population increases and dietary changes are requiring higher P inputs in cultivated land and increased mining of P ores, and human dietary shifts may have been responsible for half of the increase of P ore mining. It can mitigate regional P imbalances in agricultural soils, by optimizing phosphate fertilizer application and recycling P. Again, special thanks to you for your good comment and correction in our manuscript, which are all valuable and very helpful for revising and improving our paper. We have studied comments carefully and have made correction and improvement according to the reviewer's suggestion. These changes will not influence the content and framework of the paper. We appreciate for Editors/Reviewers' warm work earnestly. We hope that the correction will meet with approval for publication, and look forward to hearing from you at your recent convenience.

Yours sincerely, Fei Lun on behalf of all co-authors

Please also note the supplement to this comment:
https://www.earth-syst-sci-data-discuss.net/essd-2017-41/essd-2017-41-AC1-supplement.pdf

---

## Author Comment (AC2) · 17 Oct 2017

Dear Editors and Reviewers: Thank you for your letter and for the reviewers' comments concerning our manuscript entitled "Global and regional phosphorus budgets in agricultural systems and their implications for phosphorus-use efficiency". Those comments are all valuable and very helpful for revising and improving our paper, as well as the important guiding significance to our researches. We have studied comments carefully and have already made the revision. The revised portions are marked in red in this reply. The main corrections in the paper and the responds to the reviewer's comments are as flowing:

[Figure]

Reviewer 2: First of all, thank you so much for your great comments, and I have revised all of them as follows: Comment 1: There is also supporting text which provides additional information on sources, references and calculations. I did not find many anomalies (no fertilizer for Belgium? and a cell ref. issue in Detailed worksheet). Reply: Thank you so much for your detailed check and we are sorry for our faults. We have double-checked the data and we have found the related data for Belgium in our revised Data. Besides, we have also revised the related texts, table and figures in the revised manuscript.

Comment 2: The data in the worksheets are easy to follow. However, there are no units specified on any of the columns which significantly detracted from ease of use. Also, the sources of the data is not readily at hand, this would be very helpful. For example in Table SI-4 FAO is cited as a source of a great deal of data, but the exact reference is not given or even the year of the publication. Is it FAO 2002 cited in the reference? In fact the vast majority of the data is from FAO. Another sources, for the 'Phosphate acid' is the IFA- but again publication is not given and there is no citation. The other source is an atmospheric model reported in Liu et al 2007 but here the citation is not given on the table although it is in the text. Reply: Unit: we have added the units for all columns. For the country level, the units for cropland and pasture soil P balance per area are "kg P/ha", the remaining units are all "t P" (ton of P); for the regional level, the units for cropland and pasture soil P balance per area are "kg P/ha", the remaining units are all "Mt P" (million ton of P). Data source: in our last manuscript, all the data source are presented in the texts. For our revised manuscript, we have also added the data source below the tables, like Table SI-3 (Line 396-Line 410, Page 23 in Supporting Information) and Table SI-4 (Line 414- Line 432, Page 32 in Supporting Information).

Comments 3: Some of the parameters, too, are curious. For example, the P:N ratios and P contents in SI-3 have no source and it is curious, for example, that the values for dairy and nondairy cattle is the same. It is not clear why a range is given for P:N and not for P- surely there is uncertainty also for P even if less than P:N. Further, are the same

values used for the entire world. And is the P content based on as is or dry weight? Reply: We have also added the sources for the N and P contents of livestock and their manure, including ASAE (2005), COMIFER (2007), Levington Agriculture (1997), MWPS-18 (1985), OECD Secretariat (1991) and Sheldrick et al (2003). Then we calculated the P:N ratio for livestock manure. Due to the limited data and previous results, we can only use the same result for "dairy cattle and non-dairy cattle". Besides, all the values are the same for the entire world. The P:N ratios is used to calculate the amount of P in livestock manure. In the FAO statistics (http://www.fao.org/faostat/en/#data), they have presented the amount of N in livestock manure and their distribution. Therefore, using the P:N ratios in livestock manure, we estimated the amount of P in livestock manure and their distribution. Besides, P contents of livestock and their products referred to the ration P in livestock and their products. With the amounts of meat, eggs and milk, we calculated how much P stocked in meat, eggs and milk.

Comments 4: Table SI-5 is a summary for cropland P inputs and outputs. There are ranges given but it is not clear how these were determined. Also, I do not know what recycled crop residue means. Is it crop residue left in the field? I find the values for deposition (atmospheric) for P unexpectedly high. I realize these values have been published and were derived form a model. It would help if the source of the deposition was quantified and presented as a part of the budget, otherwise it looks like it came from outside the boundaries of the model which does not seem correct. The human sewage value also seems high. I tracked the values to Liu et al 2008, which supposed that 30% of urban sewage and 70% of rural sewage was applied to agricultural land. But When I looked at some western countries the ratio was less than 10% which seems more plausible but where did these estimates come from. In any case it is not transparent. Reply: The results for Table SI-5 is a summary of the cropland P inputs and outputs from the previous study results, and the ranges present their differences between these results. For example, fertilizer inputs ranged 13.7-15.0 Tg P yr-1 from Liu et al (2008), Smil (2000) and Cordell et al (2009), which means we summarized their results and these results were between 13.7 and 15.0 Tg P yr-1 in the year of 2000.

For the recycled crop residue, it referred to the crop residue left in the field. For the atmospheric decomposition and human sewage, our estimates were presented in Figure 2. The ranges of these two sections in the Table SI-5 were summarized from other previous studies, including Liu et al (2008), Smil (2000), Cordell et al (2009), MacDonald et al (2011), Bouwman et al( 2009, 2011). Due to the limited data, we also used Liu's results (2008) in our estimate, which supposed that 30% of urban sewage and 70% of rural sewage was applied to agricultural land for all countries in the world. The human manure to cropland amounted to 1.37 Tg P yr-1 for the period of 2002~2010, which was in the range of 1.3-1.5 of other previous studies (Liu et al., 2008; Cordell et al., 2009). However, we used the same level for all countries in the world and there were no differences among countries, and therefore to do further detailed research on human sewage to croplands and pastures in our future next research. Comment 5: In Table SI-7 cropland PUE> total PUE; this seems wrong and is in contrast to lines 271-272. Reply: We have corrected the Table SI-7 in the revised Supporting Information (see Line 452-453, Table SI-7, Page 29 in Supporting Information).

Comments 6: Globally, half of the total P input (21.3 TgP yr-1) into agricultural systems accumulated in agricultural soils during this period, with the rest lost to bodies of water through complex flows. Comment I could not find this in the document Reply: In our manuscript, this results presented in our manuscript "Line 268-270" as follows: "Outputs from the agriculture system amounted to 12.5 Tg P yr-1, which combines outputs from leaching and runoff into bodies of water (5.4), non-recycled manure waste (4.3) and sewage (2.2), bio-energy (0.4), and burned crop residues (0.2)" .

Comments 6: Global P accumulation in agricultural soil increased from 2002 to 2010, despite decreases in 2008 and 2009, and the P accumulation occurred primarily in cropland. Despite the global increase of soil P, 32% of the world's cropland and 43% of the pasture had soil P deficits. Comment: I could not find this in the document Reply: In our manuscript, this results presented in our manuscript "Line 289-291", "Figure 3", "Line 360-367", and "Line 367-369" as follows: Overall, P in agricultural
soils increased by 1.3% annually for the period of 2002~2010, whereas P losses to the environment increased faster (6.4% yr-1) than fertilizer inputs. About 32% of the global cropland area (in 75 countries) had annual soil P deficits from 2002 to 2007, with the net cropland soil P accumulation of 6.20~7.66 Tg P yr-1. This fraction increased to 50% in 2008 and 2009 but the net cropland soil P accumulation decreased to 4.38Tg P yr-1 in 2008 and 5.39 Tg P yr-1, at the time of the global financial crisis, as a result of high P prices and the resulting reduction in fertilizer application (Cordell et al. 2009, 2012). However the fraction of cropland soil P deficits returned close to the decadal mean value in 2010, with the net soil P accumulation of 7.30 Tg P yr-1. Compared with croplands, a slightly larger proportion of the total global pasture area had a net annual soil P deficit from 2002 to 2010, mostly in Europe and North America. The deficit proportion of grassland was only about 38% in 2002 and 2003, with the annual net soil P accumulation of 2.26 Tg P yr-1; however, it increased to 43% during the period of 2004~2010 and the annual pasture soil P accumulation was about 2.10 Tg P yr-1, with the smallest of 2.00 Tg P yr-1 in 2009.

Comments 7: The trend of increasing consumption of livestock products will require more P inputs to the agricultural system, implying a low P-use efficiency aggravating the P stocks scarcity in the future. Comment: While technically this is correct, I don't really see a strong connection with the data set. There is more prospects for recycling animal P than N. Reply: Global pasture soil P balance mask large regional differences and global pasture soil P deficit mostly occurred in Europe and North America. These countries export livestock products, and only a small fraction of livestock manure is recycled to pasture, so there is currently a soil P deficit; in the long term, this may result in a loss of soil fertility. Consequently, improving the manure utilization efficiency and applying more livestock manure to pasture will be important strategies (Wu et al., 2014). Dietary changes to more livestock products are requiring higher P inputs in cultivated land and increased mining of P ores (Grote et al., 2005; Foley et al., 2011), and human dietary shifts may have been responsible for half of the increase of P ore mining. Besides, livestock PUE were much lower than other PUEs, and it is very

important to improve livestock PUE. Different from N, P ores are the limited sources, and it should be paid more attentions on livestock P.

Again, special thanks to you for your good comment and correction in our manuscript, which are all valuable and very helpful for revising and improving our paper. We have studied comments carefully and have made correction and improvement according to the reviewer's suggestion. These changes will not influence the content and framework of the paper. We appreciate for Editors/Reviewers' warm work earnestly. We hope that the correction will meet with approval for publication, and look forward to hearing from you at your recent convenience.

Yours sincerely, Fei Lun on behalf of all co-authors

Please also note the supplement to this comment:
https://www.earth-syst-sci-data-discuss.net/essd-2017-41/essd-2017-41-AC2-supplement.zip